# Prognostic Value of Lymph Node Ratio in Patients with Uterine Carcinosarcoma

**DOI:** 10.3390/jpm14020155

**Published:** 2024-01-30

**Authors:** Rasiah Bharathan, Stephan Polterauer, Martha C. Lopez-Sanclemente, Hanna Trukhan, Andrei Pletnev, Angel G. Heredia, Maria M. Gil, Irina Bakinovskaya, Alena Dalamanava, Margarita Romeo, Dzmitry Rovski, Laura Baquedano, Luis Chiva, Richard Schwameis, Ignacio Zapardiel

**Affiliations:** 1Department of Obstetrics and Gynecology, Medical University Vienna, 1090 Vienna, Austria; rasiah.bharathan@meduniwien.ac.at (R.B.);; 2Hospital de Torrecárdenas, 04009 Almeria, Spain; marthalo90@gmail.com; 3N.N. Alexandrov National Cancer Center, 223040 Minsk, Belaruselen-d@tut.by (A.D.);; 4Clinica de Especialidades de la Mujer, Mexico City 03810, Mexico; 5Gynecologic Oncology Unit, La Paz University Hospital, 28046 Madrid, Spain; mmar1984@hotmail.com (M.M.G.);; 6Instituto Catalan de Oncologia Badalona, 08916 Barcelona, Spain; mromeo@iconcologia.net; 7Hospital Universitario Miguel Servet, 50001 Zaragoza, Spain; 8Obstetrics and Gynecology Department, Clinica Universidad de Navarra, 28027 Madrid, Spain; lchiva@unav.es

**Keywords:** uterine carcinosarcoma, lymph node ratio, prognosis

## Abstract

Uterine carcinosarcoma is a rare high-grade endometrial cancer. Controversy has surrounded a number of aspects in the diagnosis and management of this unique clinicopathological entity, including the efficacy of adjuvant therapy, which has been questioned. An unusual surgico-pathological parameter with prognostic significance in a number of tumour sites is the lymph node ratio (LNR). The availability of data in this respect has been scarce in the literature. The primary aim of this collaborative study was to evaluate the prognostic value of LNR in patients with uterine carcinosarcoma. LNR is a recognized lymph node metric used to stratify prognosis in a variety of malignancies. In this European multinational retrospective study, 93 women with uterine carcinosarcoma were included in the final analysis. We used *t*-tests and ANOVA for comparison between quantitative variables between the groups, and chi-square tests for qualitative variables. A multivariate analysis using Cox regression analysis was performed to determine potential prognostic factors, including the LNR. Patients were grouped with respect to LNR in terms of 0%, 20% > 0% and >20%. The analysis revealed LNR to be a significant predictor of progression-free survival (HR 1.69, CI (1.12–2.55), *p* = 0.012) and overall survival (HR 1.71, CI (1.07–2.7), *p* = 0.024). However, LNR did not remain a significant prognostic factor on multivariate analysis. Due to limitations of the retrospective study, a prospective large multinational study, which takes into effect the most recent changes to clinical practice, is warranted to elucidate the value of the pathophysiological metrics of the lymphatic system associated with prognosis.

## 1. Introduction

Carcinosarcoma was previously known as malignant mixed Mullerian tumour (MMMT) and can affect the uterus, ovary or the cervix. Uterine carcinosarcoma (UCS) is a distinct metaplastic subtype of high-grade endometrial cancer [1]. This distinction realigns this entity from the earlier classification as a uterine sarcoma to that of a high-grade endometrial malignancy. UCS is a rare gynaecologic cancer, which constitutes 2−8% of all uterine malignancies [2,3]. The aggressive variant of carcinosarcoma of this heterogenous group has an estimated 5-year overall survival (OS) ranging from 33% to 39%. The international incidence of this malignancy is estimated to be 0.5–3.3 per 100,000, with a steady increase in the incidence. The factors that are thought to influence this apparent rise in the incidence include lengthening life expectancy, potentially better understanding of the genesis of the disease amongst pathologists, as well as the development and application of better analytical techniques in the laboratory.

UCS is a fascinating tumour, regarded as a biphasic tumour with the co-existence of features characteristic of carcinomatous (epithelial) and sarcomatous (mesenchymal) elements [4]. The epithelial component could be endometrioid, serous, clear cell, mucinous, squamous or undifferentiated. The mesenchymal elements could have derived from epithelial origin (due to metaplasia or trans-differentiation) and these could either be homologous (endometrial stromal sarcoma, leiomyosarcoma, fibrosarcoma and undifferentiated sarcoma) or heterologous (rhabdomyosarcoma, osteosarcoma, chondrosarcoma and liposarcoma) in terms of its histological characteristics [5]. In a recent study, Sertier et al. strengthened the theory that both epithelial and mesenchymal components are derived through clonal evolution and transcriptomic reprogramming [6]. Indeed, the presence of heterologous components is associated with a worse prognosis [7].

UCS tends to be diagnosed at a more advanced stage than the other high-grade endometrial cancers [4]. The stage at diagnosis of UCS appears to demonstrate a bimodal distribution, with 40–50% of women at an early stage (International Federation of Gynecology and Obstetrics ((FIGO) I–II) and 50–60% at an advanced stage (FIGO III–IV) [8]. The management of UCS has been controversial. Multimodal therapy offers the best prognosis, with surgery as the primary treatment in suitable patients. Because the genesis of UCS is now understood to be a process of trans-differentiation (conversion theory), the management of UCS is aligned with that of high-grade endometrial carcinoma, as recognized in international guidelines [8,9,10,11].

Therefore, standard surgical staging of uterine carcinosarcoma presumed to be confined to the uterine corpus incorporates total hysterectomy, bilateral salpingo-oophorectomy and evaluation of the lymph nodes (LN), either by sentinel node biopsy or full lymphadenectomy (in the pelvic and para-aortic regions), is recommended [8,10]. In the 50–60% of patients who present with advanced-stage disease, surgical debulking, including the removal of suspicious lymph nodes, is recommended, with the aim of complete tumour resection.

In women with subtypes of high-grade disease, the value of adjuvant therapy has demonstrated a diverse trend in terms of survival benefit. In a recent systematic review of the impact of adjuvant therapy, a possible trend towards the benefit of adjuvant therapy was noted; however, a statistically significant effect could not been reached in the pooled analysis of patients with early-stage disease [12]. However, in those with advanced disease, adjuvant therapy has demonstrated a clear survival benefit [13].

In a large Japanese multicentre retrospective study, older age, residual disease at surgery, the finding of a large tumour, dominance of sarcomatous features, deep myometrial invasion, lymphovascular space invasion and advanced-stage disease were all independently associated with decreased PFS (*p* < 0.01) [5]. In addition, LN positivity appears to serve as an adverse prognostic marker [14].

The concept of the lymph node ratio (LNR) has proven to be of prognostic value in multiple gynaecological tumours, encompassing vulval, cervical and ovarian malignancies, with regards to progression-free survival (PFS) or OS [15,16,17]. LNR or lymph node density (LND) is regarded as an index of disease burden when the lymph node count is also incorporated. Indeed, LNR remains the most reliable indicator of disease burden within the lymphatic system. LNR has been demonstrated to be of prognostic value in advanced-stage endometrial cancer [18,19]; however, there is a paucity of knowledge regarding the potential prognostic value of LNR in the setting of uterine carcinosarcoma [20].

The aim of the present study was to evaluate the impact of LNR on the oncological outcomes of uterine carcinosarcoma in terms of progression-free survival and overall survival.

## 2. Materials and Methods

The SARCUT study was a retrospective study; this was a pan-European international collaboration. The study captured data from 966 patients diagnosed with uterine sarcoma and carcinosarcoma from January 2001 to December 2007, with a 5-year minimum follow-up period, until 2012. As the study coordinating centre, ethics approval for this study was obtained from La Paz University Hospital ethics committee (#PI-1382) (P^o^ de la Castellana, 261, 28046 Madrid, Spain). Each participating centre had to comply with the local guidelines with respect to project classification and seeking necessary approval from the institutional ethics committee. Interested participants were invited to contribute to this project via online calls, which were disseminated through national and European societies and research platforms within the gynaecological oncology community. An example of this communication portal is the European Network of Young Gynae Oncologists (ENYGO). A total of 46 gynaecological oncology departments and 53 researchers contributed to this international effort. Patients who underwent primary treatment in other centres, patients with incomplete data regarding recurrence or survival, as well as patients without at least one node removed from the pelvic or para-aortic areas were excluded. The same oncological team in each centre performed all surgical interventions and follow-up of the patients. The decision regarding adjuvant therapy followed appropriate national guidelines, which could have been tailored to individual patients by the local multidisciplinary tumour board.

A web-based encrypted database was employed to capture the data from each patient and each centre was assigned a unique identification code. The individual researcher was responsible for capturing and entering all relevant information pertaining to each patient. All data regarding the medical history, diagnosis, staging according to the FIGO 2009 classification [21], management and follow-up was recorded in the database. The surgical approach was determined by individual centres and the variability of the extent of cytoreductive surgery was partly a function of the surgical team. Following surgical treatment, the follow-up comprised of clinical evaluation, relevant blood tests and regular cross-sectional imaging (e.g., computed tomography [CT] or magnetic resonance imaging [MRI]). Uniform nomenclature for surgeries performed, pathologic details and follow-up was used. The classification of surgical cytoreduction was performed according to Zapardiel and Morrow classifications [22]. Local recurrence was defined as the appearance of the tumour in the same location after a minimum disease-free period of 6 months, and distant recurrence when it appeared in a new location after treatment. The diagnosis of recurrence was accepted based on either tissue biopsy with histological confirmation or a corresponding radiological constellation. The follow-up schedule mandated an appointment every 3 months during the first year of follow-up and then appointments every 6 months for a period of up to 5 years after treatment. Subsequent follow-up took place annually for a further 5 years.

LNR was defined as the ratio of the number of positive LNs to the number of removed LNs for a given lymph node basin (either a pelvic or para-aortic lymph node basin). The pelvic nodal basin was not separated into unilateral subunits. Patients were stratified into three risk groups according to LNR (0 vs. 0% < 20% vs. >20%), as published previously in a report from the Gynecologic Oncology Group (GOG), protocol #37 [23].

### Statistical Analysis

Quantitative variables were described by mean values (standard deviation) and qualitative data was described by absolute values and percentages. We used *t*-tests and ANOVA for comparison between quantitative variables between the groups, and chi-square tests for qualitative variables. Multivariate analysis using Cox regression analysis was performed to determine potential prognostic factors, including the LNR. Pearson correlation analyses explored associations between the LNR and other quantitative variables. Survival analysis was performed using Kaplan–Meier curves and log-rank tests. All comparisons were two-tailed, and the alpha error was set at 5%. All data were analysed using SPSS software version 29.0 (SPSS Inc, Chicago, IL, USA).

## 3. Results

In the original SARCUT project, a total of 966 patients were recruited from across all the centres in this European multicentre collaboration. A subgroup of this cohort comprised of 283 patients with uterine carcinosarcoma. From this group, a cohort of 93 patients (32.9%) were considered eligible for this particular analysis, with all the relevant information to calculate LNR. Exclusion was due to the LND not being performed, only one LN being resected or because the number of resected nodes were not provided. Table 1 summarises the patient characteristics, treatment strategies and follow-up status. The mean ± standard deviation follow-up time was 32.5 ± 36.2 months.

The median (IQR) number of resected and positive lymph nodes from the pelvis was 14 (6–27) and 0 (0–1), respectively. The median (IQR) number of resected and positive lymph nodes from the para-aortic region was 6.5 (3.5–10.5) and 0 (0–0), respectively. The median (IQR) number of total resected LNs in LNR groups 0%, 0% < 20% and >20% were 14 (6–28), 31 (17–42) and 6.5 (3–11), respectively.

LNR demonstrated a moderately significant correlation (r = 0.47, *p* < 0.001) with FIGO status and a weak association with LVSI (r = 0.29, *p* = 0.02). LNR was not associated with tumour size.

Figure 1 displays the distribution of the number of pelvic lymph nodes resected. The graph displays metrics in relation to pelvic nodes only, as only the pelvic group demonstrated prognostic value. A total of 21 (26%) and 7 (18%) patients were identified as having positive pelvic and para-aortic nodes, respectively. Patients were stratified into three risk categories according to LNR values: 0% (n = 69, 74.2%), 0% < 20% (n = 8, 8.6%) and >20% (n = 16, 17.2%), as previously described (Kunos 2009).

In the univariate analysis, FIGO stage was a significant predictor of PFS (*p* = 0.021). In the univariate analysis, the following factors demonstrated significant prognostication for PFS: LNR (*p* = 0.012), residual disease (*p* ≤ 0.001), tumour size (*p* = 0.034) and positive pelvic LNs (*p* = 0.013). Similarly, the following factors were significant predictors of OS in the univariate analysis: LNR (*p* = 0.024), residual disease (*p* = 0.01), tumour size (*p* = 0.049) and positive pelvic LNs (*p* = 0.005).

A multivariate analysis to identify independent predictors was performed. In this multivariate analysis, tumour size remained a significant predictor of PFS (*p* = 0.045) and OS (*p* = 0.38). In addition, positive margin was also a predictor of PFS (*p* = 0.015) (Table 2 and Table 3). Figure 2 illustrates the PFS and OS with regard to LNR using the Kaplan–Meyer curve.

## 4. Discussion

This international retrospective study investigated the value of LNR in predicting the PFS and OS in women with UCS. UCS, which is a variant of endometrial carcinoma, is associated with a more adverse outcome [24]. Our study identified LNR as a significant predictor of PFS (*p* = 0.012) and OS (*p* = 0.024). However, LNR did not serve as an independent predictor of PFS or OS. A recent database study by Gao et al. identified LNR as a significant predictor of PFS and OS in UCS, but as in our study, it did not remain a significant factor in multivariate analysis [20]. The index study confirms the findings of the Surveillance, Epidemiology and End Results (SEER) database analysis by Gao et al. in that FIGO stage, tumour size and LN positivity are also significant predictors of PFS and OS. Unlike these findings, LNR in endometrial cancer is proven to be an independent predictor of OS [19,25].

Lymph node involvement in uterine sarcoma leads to adverse survival outcomes [26]. The concept of LNR appears to hold prognostic value in several tumours, including gynaecological cancers [15,16,17]. This concept indicates LN involvement but not the full extent of disease burden within the LN basin because it is a ratio of resected LNs not an index of surgical radicality. LNR was of prognostic value, for instance when the mean number of resected LNs was found to be variable, in cervical cancer with values of 20 and 19 [27,28] and in vulval cancer with values of 14 [29], 15 [15] and 10 [30]. Indeed, in endometrial cancer, the median number of nodes resected was 17 in the pelvic region and 6 in the para-aortic region [18], 11 in the para-aortic region in another study [19] and yet another study reported 13 in the pelvic region and 7 in the para-aortic region [31]. In the present study, the mean pelvic LN count was 15 and the mean para-aortic count was 8, which is in keeping with the reported range in the literature. Therefore, the extent of LN resection in the present study is unlikely to account for the absence of LNR as a significant prognostic factor. Although LN involvement in uterine carcinosarcoma is controversial, incomplete cytoreduction and residual disease (including nodal disease) are amongst the most important prognostic factors [26].

An aspect of the pathology of UCS that might bear influence in our analysis is the biphasic nature of this interesting entity. The epithelial and mesenchymal components are believed to be of significance in determining prognosis [5,6]. Several studies have hinted that the epithelial component may be the dominant factor in determining prognosis [5,14,32]. However, in relation to the characteristics of the mesenchymal component, heterogenous histology compared to homogenous histology appears to render the oncological outcomes worse [7].

A recent single-centre study comparing SLN-alone versus lymph node dissection in uterine carcinosarcoma revealed no OS difference at three years in a cohort of patients receiving adjuvant therapy [33]; thus, further investigation of the lymphatic system’s significance in uterine carcinosarcoma is essential. This finding would suggest that in those who have evidence of metastasis, regardless of how the diagnosis was achieved, will garner survival benefit from adjuvant therapy. There may be several explanations as to the non-significant prognostication with LNR.

LN positivity and the number of affected nodes has been demonstrated to have a prognostic value in uterine sarcoma, including CS [26]. The absence of independent prognostic association with LNR (or LND) may simply reflect the aggressive nature of sarcomatous disease. This would suggest that any indication of LN involvement is a poor prognostic factor, regardless of the true extent of LN metastasis. This appears to lend support to the finding that SLN biopsy alone for apparent early-stage CS has a comparable survival outcome to lymphadenectomy in CS [33]. To the best of our knowledge, no studies have documented the relevance of LNR in relation to sarcomatous disease of any other tumour sites. This may simply reflect the ‘negative study’ bias in scientific publication or that tumour biology in sarcomatous disease predominates disease burden in relation to LNR metrics.

UCS is a high-grade tumour. The genetics of locoregional LN metastasis is thought to differ from that of distant metastasis [34]. Local metastasis appears to have more heterogenous clonal characteristics compared to distant metastases [34]. This may also have an unverified influence on the dissemination pattern of this biphasic tumour.

Further, potential explanations with respect to the absence of LNR prognostic power may be related to the fact that midline organs have a complex bilateral drainage pattern. This means that adequate harvesting of LNs from both sides would be required to ensure a reliable LNR.

A crucial aspect underpinning the use of LNR as a metric is the accurate evaluation of surgical specimens and stringent study of the LNs. Several factors have been identified as influencers of the LN count. These include anatomical variation, tumour biology, experience and expertise of the pathologist, as well as the processing chemicals used in preparing the tissue blocks [35,36]. Others include the well-recognised technical aspect of tissue processing in the laboratory, utilisation of standard protocols across individual laboratories, the recognition of nodal structures and reporting. This has been articulated in the literature [37]. Sherbeck and colleagues have demonstrated the variations seen in LN counting, even amongst an international cohort of pathologists [37]. In fact, the intra-observer and inter-observer variability in determining the LN count has also been previously documented by Prakash and colleagues [38]. Therefore, the current study might have suffered from the fact that, although it is an international study, paradoxically, wide collaboration might also be a limiting factor.

One additional factor, which may be of importance, is age-related changes in lymph node count and function. Age-related decline in LN count and altered lymph dynamics are recognized features of immunosenescence [39,40]. Ahmadi and colleagues, in their systematic review, described a range of changes that are recognized to correlate with the aging process. These include fibrosis, hyalinization, fat deposition and the appearance of ‘spaces’ within lymph nodes [39]. Further changes that have been noted include a reduction in the cortex and the medulla of lymphoid tissue, as well as a decline in the number and size of germinal centres and the expansion of medullary sinuses [39]. The summative effect of these changes is altered lymphatic flow, which may in turn have a bearing on LNR. This may be a relevant fact given that carcinosarcoma of the uterus is predominantly a disease of the older patient.

In the current cohort, positive pelvic LNs, residual disease and LNR have all been proven to be significant predictors of PFS and OS. However, LNR did not remain a significant factor in multivariate analysis. Despite this, LNR could be considered a factor in the post-treatment follow-up strategy of the patients.

There are a number of limitations in this study, which should be recognized by the reader. This study was retrospective in nature. The limited sample size from a large multinational study, where variations in the extent of surgery and adjuvant therapy, will inevitably bear influence. Indeed, the robustness of histological evaluation may also contribute to this finding. In this regard, further data regarding histological components would be essential in delineating the role of lymphatic pathophysiology on oncological outcomes. We did not have this information available. Evidence from the literature suggests that age, for instance, determines the likelihood of a patient receiving adjuvant therapy [41]. Indeed, the quality of staging appears to correlate with prognosis [42]. This could manifest here as a type II error.

LNR is regarded as an indicator of tumour load in the lymphatic system. In this regard, in the future we could seek alternative methods to characterise and quantify the tumour load deposited within the lymphatic system. This could encompass a range of metrics, from serum biomarkers that indicate lymphovascular space invasion and molecular signatures of tumours to radiomics that depict tumour volumes within the lymphatic system. An illustrative example of the latter would be ultrasmall particles of iron oxide (USPIO) in imaging lymph node metastasis [43]. In this study the radiologist can visualize negative contrasting of the metastasis [43]. The application of artificial intelligence to such a scenario could yield a volumetric evaluation of lymph node burden. As an alternative, one could combine this with biomarkers to derive composite metrics of tumour load. Indeed, with the fusion of imaging techniques, the development of optical probes, which are disease process specific, and data processing algorithms could offer new opportunities for assessing tumour load of the lymphatic system, which are less invasive from the patient’s perspective. Particularly, when the optical probes are grounded on prognostically validated markers and delivered using nanoparticles as a vehicle, one can anticipate a truly compelling paradigm shift in stratifying patients’ cancer biology.

## 5. Conclusions

Uterine carcinosarcoma is a rare but highly aggressive uterine malignancy. LNR could potentially offer additional weight in treatment planning. In addition, due to the rapidly evolving understanding of the genetics and epigenetics of the genesis of this heterogenous entity, a large international prospective effort would be required. Alongside this evolution in the understanding of the disease process, clinical management guidance has also changed, with the rapid implementation of sentinel lymph nodes in evaluating lymph node metastasis. We will need to redefine clinically meaningful pathophysiological metrics in relation to the lymphatic system to tailor adjuvant therapy. Further retrospective work can also yield relevant information to re-evaluate the role of LNR in the context of molecular signatures to refine individual treatment protocols.

## Figures and Tables

**Figure 1 jpm-14-00155-f001:**
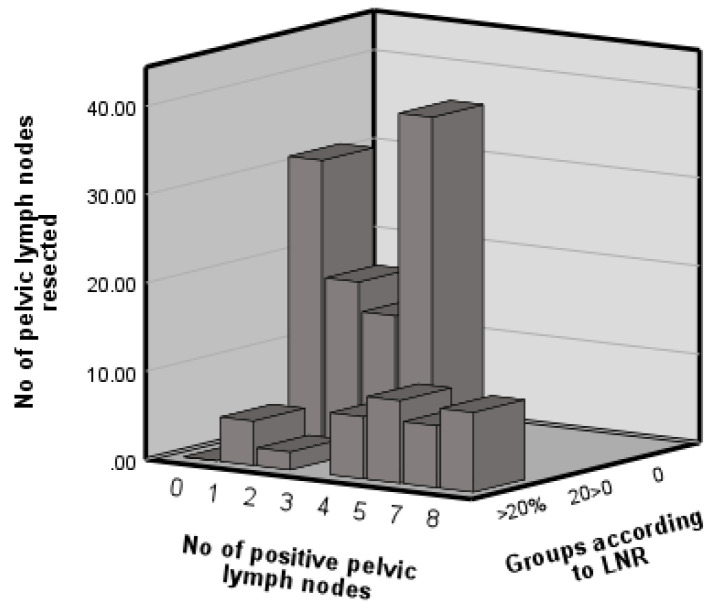
Distribution of pelvic lymph nodes resected and involved and the lymph node ratio.

**Figure 2 jpm-14-00155-f002:**
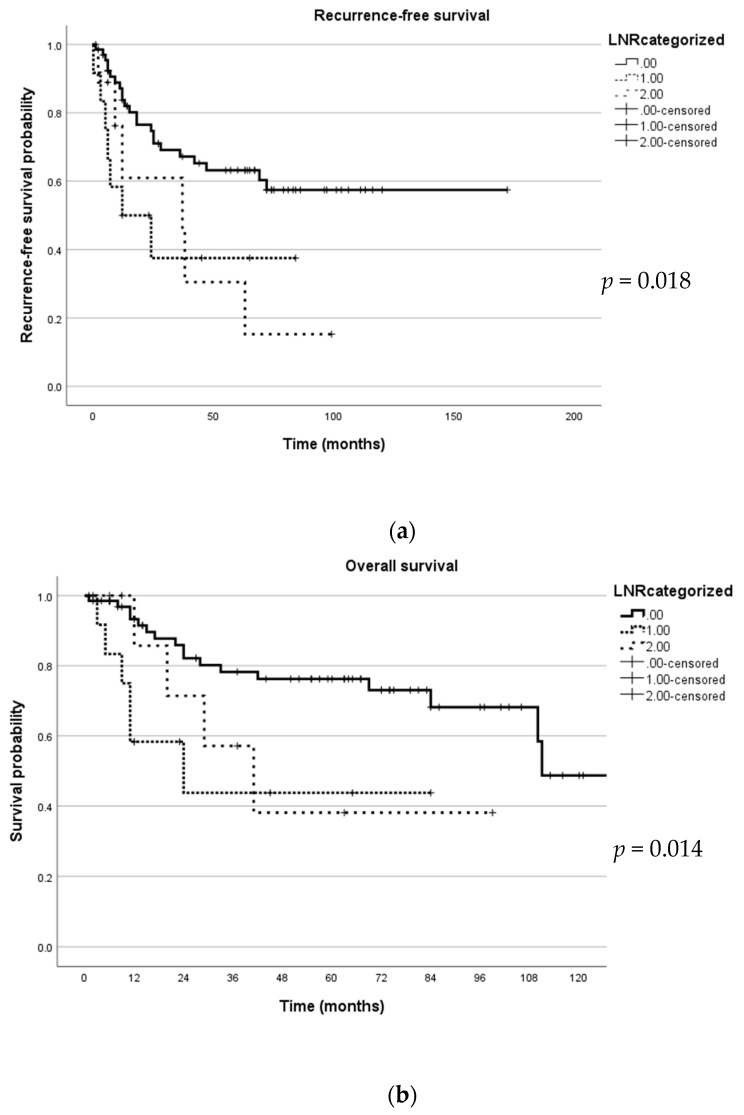
Kaplan–Meyer curve depicting survival in relation to lymph node ratio (LNR). (**a**) This figure illustrates recurrence-free survival; (**b**) this figure illustrates overall survival.

**Table 1 jpm-14-00155-t001:** Characteristics of patients (n = 93). SD: standard deviation; LNR: lymph node ratio; LVSI: lymph vascular space invasion; cm: centimetres; n: number of patients.

Characteristics	Distribution
Age (years), mean (SD)	65 (8.9)
Menopause	81 (87.1%)
Symptoms	
Pain	4 (4.3%)
Pelvic mass	4 (4.3%)
Bleeding	78 (83.9%)
Other	7 (7.5%)
FIGO Stage	
I	58 (62.4%)
II	3 (3.2%)
III	25 (26.9%)
IV	7 (7.5%)
Route of Surgery	
Laparoscopy	4 (4.3%)
Laparotomy	89 (95.7%)
Lymphadenectomy	
Pelvic only	53 (57%)
Para-aortic only	12 (12.9%)
Both	28 (30.1%)
Residual disease	
Complete resection	71 (76.3%)
Minimal residual disease (<1 cm)	4 (4.3%)
Gross residual disease (>1 cm)	2 (2.2%)
Not available	16 (17.2%)
Lymph node positivity	
Pelvic only	17 (18.3%)
Para-aortic only	3 (3.2%)
Both	4 (4.3%)
Lymph node count	
Pelvic, mean (SD)	15 (13.4)
Para-aortic, mean (SD)	8.2 (6.5)
LNR groups	
0%	69 (74.2%)
0% < 20%	8 (8.6%)
≥20%	16 (17.2%)
LVSI	
Negative	40 (43%)
Positive	25 (26.9%)
Not available	28 (30.1%)
Necrosis	
Yes	24 (25.8%)
No	32 (34.4%)
Not available	37 (39.8%)
Adjuvant therapy	
Chemotherapy	15 (16.1%)
Radiotherapy	33 (35.5%)
Both	19 (20.4%)
Recurrence	38 (40.9%)
Disease-free survival, mean (SD) months	42.7 (38.1)
Overall survival, mean (SD) months	46.9 (38.3)

**Table 2 jpm-14-00155-t002:** Characteristics influencing progression-free survival. LNR: lymph node ratio; LVSI: lymph vascular space invasion; LNs: lymph nodes; HR: hazard ratio; CI: confidence interval.

Variables	Univariate Analysis			Multivariate Analysis		
	HR	95% CI	*p* Value	HR	95% CI	*p* Value
Age	1.82	0.93–3.55	0.08			
FIGO Stage	1.41	1.05–1.89	0.021	1.39	0.99–1.96	0.058
Adjuvant radiotherapy	0.98	0.51–1.9	0.95			
Chemotherapy	1.84	0.94–3.58	0.07			
LNR	1.69	1.12–2.55	0.012	1.51	0.6–3.78	0.38
LVSI	1.66	0.79–3.46	0.18			
Positive margin	3.75	1.55–9.08	0.003	4.63	1.35–15.93	0.015
Residual disease	7.34	2.5–21.56	<0.001	3.79	0.76–18.85	0.103
Tumour size > 5 cm	2.11	1.06–4.23	0.034	2.33	1.02–5.32	0.045
Positive pelvic LNs	1.25	1.05–1.5	0.013	0.91	0.56–1.46	0.69
Surgical approach	1.68	0.23–12.28	0.61			

**Table 3 jpm-14-00155-t003:** Characteristics influencing overall survival. LNR: lymph node ratio; LVSI: lymph vascular space invasion; LNs: lymph nodes; HR: hazard ratio; CI: confidence interval.

Variables	Univariate Analysis			Multivariate Analysis		
	HR	95% CI	*p* Value	HR	95% CI	*p* Value
Age	1.95	0.9–4.2	0.09			
FIGO Stage	1.14	0.8–1.64	0.47			
Adjuvant radiotherapy	1.21	0.55–2.64	0.64			
Chemotherapy	1.29	0.58–2.89	0.54			
LNR	1.71	1.07–2.7	0.024	1.4	0.51–3.82	0.52
LVSI	0.88	0.37–2.03	0.76			
Positive margin	2.09	0.72–6.09	0.18			
Residual disease	2.94	1.23–6.68	0.01	1.6	0.39–6.6	0.52
Tumour size > 5 cm	2.14	1.00–4.54	0.049	1.13	1.05–5.96	0.038
Positive pelvic LNs	1.33	1.09–1.62	0.005	1.13	0.69–1.84	0.64
Surgical approach	1.15	0.16–8.57	0.89			

## Data Availability

The data presented in this study are available on request from the corresponding author. Due to data privacy issues, the data is not available on a public repository.

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
