# Peer review of "Prognostic Value of Lymph Node Ratio in Patients with Uterine Carcinosarcoma"

_jpm, 2024, doi:10.3390/jpm14020155_

Round 1
Reviewer 1 Report
Comments and Suggestions for Authors
This is a retrospective study on the prognostic value of lymph node ratio in patients with uterine carcinosarcoma.
The manuscript is not written properly.
The abstract and title sufficiently reflect the content.
The keywords are sufficient.
The introduction provides sufficient background.
The methods are not properly described. The material and methods should be expanded.
The reference style is not correct. Citations have not been used properly.
The presentation of the work is not clear, with regards to language and grammar.
Abbreviations are not used properly. There are multiple errors and the text needs major editing.
Abbreviations used in the text should be checked and the manuscript needs editing.
Citation style ‘(Harano 2016)’ should be corrected in line 52.
The material and methods section should be expanded.
Citation style ‘(Goa 2021)’ should be corrected in line 147.
I think it should be re-evaualuated after the corrections.
Comments on the Quality of English LanguageEditing of English language is required.
Author Response
I would like to thank the reviewer for the detailed comments and for considering my responses in this revision.
Abbreviations used in the text should be checked and the manuscript needs editing. This has been checked and improved as appropriate.
Citation style ‘(Harano 2016)’ should be corrected in line 52. This has been amended.
The material and methods section should be expanded. This has been expanded.
Citation style ‘(Goa 2021)’ should be corrected in line 147. This has been corrected.
The reference style is not correct. Citations have not been used properly. In our original submission we had used the MDPI style in Endnote; this widget for this particular style had been downloaded from teh website. However, i have now removed the doi data at the end of the reference. I hope that meets the requirements.
The presentation of the work is not clear, with regards to language and grammar. This aspect has been improved. However, if there are further grammatical errors, please help us by providing exmaples so that we can improve further.
Reviewer 2 Report
Comments and Suggestions for Authors
The authors assessed the prognostic significance of LNR in 93 patients operated on for carcinosarcoma. In a univariate analysis, they showed a significant impact of the LNR value on PFS and OS. This was not confirmed by multivariate analysis. In an exhaustive discussion, they drew attention to the LNR parameter in the context of the importance of positive lymph nodes as an important element of optimal cytoreduction and an important prognostic factor. They drew attention to the possible impact of patients' age on the functioning of the lymphatic system. They also drew attention to the need to collect prospective data, which would allow for a reliable assessment of the importance of LNR in patients with carcinosarcoma.
Below are some more detailed comments
Abstract:
Lines 25-26: …”prospective evaluation is warranted to elucidate the value of LNR in shaping treatment decisions” – In my opinion, this wording requires correction. The subject of the study was to assess the prognostic, rather than predictive, significance of LNR (the predictive factor assesses the susceptibility of a given group of patients to a specific treatment method).
Material and Methods:
Lines 68-69 : …”patients without at least 1 node removed from the pelvic or para-aortic areas were excluded”- In the context of the definition of LNR (LNR was defined as the ratio of the number of positive LN to the number of removed LN), a theoretically significant number of pts in the analyzed group with a small number of removed lymph nodes may influence the assessment of the prognostic significance of this parameter. There are studies (concerning cancers of other locations) showing that LNR is a significant factor for DFS, OS, and cancer-specific survival in patients in whom more than 10 lymph nodes were resected. It is obviously that the number of positive LNs identified is dependent on the total number of LNs resected for pathological examination. I suggest adding a comment.
Results:
Lines 100-103: points out that in the group of 283 pts diagnosed with carcinosarcoma, only in 1/3 (93 pts) the available data allowed the calculation of LNR. The group excluded from further analysis most likely included patients who did not have any lymph node removed. A short comment seems appropriate at this point.
​In Table 1, a relatively important group of 16/93 pts with no information about the completeness of the surgical procedure and 28/93 about LVSI is noteworthy. How does this relate to paragraph 1 of the Material and Methods section, which states that patients with incomplete data were excluded from the analysis (line 68-69)?
Results:
Lines 118-120: …”In multivariate analysis, tumor size remained a significant predictor of PFS (P=0.045) and OS (P=0.38); in addition, 119 positive margin was also a predictor of PFS (P=0.015) (Tables 2 and 3)” - I propose to supplement table 1 with data on pts with positive margins. Judging by the number of pts: Complete resection (71) + Minimal residual disease (<1cm) (4) +Gross residual disease (>1cm) (2) + Not available (16) = 93, then pts with a positive margin have been included here somewhere ( probably to minimal residual disease). Why was this subgroup (positive margin) separated later?
Discussion:
The discussion is exhaustive and I have no comments on this part.
Author Response
Abstract:
Lines 25-26: …”prospective evaluation is warranted to elucidate the value of LNR in shaping treatment decisions” – In my opinion, this wording requires correction. The subject of the study was to assess the prognostic, rather than predictive, significance of LNR (the predictive factor assesses the susceptibility of a given group of patients to a specific treatment method). The wording has been amended to reflect this - a prospective large multinational study, which takes into effect the most recent changes to clinical practice, is warranted to elucidate the value of the pathophysiological metrics of the lymphatic system associated with prognosis
Material and Methods:
Lines 68-69 : …”patients without at least 1 node removed from the pelvic or para-aortic areas were excluded”- In the context of the definition of LNR (LNR was defined as the ratio of the number of positive LN to the number of removed LN), a theoretically significant number of pts in the analyzed group with a small number of removed lymph nodes may influence the assessment of the prognostic significance of this parameter. There are studies (concerning cancers of other locations) showing that LNR is a significant factor for DFS, OS, and cancer-specific survival in patients in whom more than 10 lymph nodes were resected. It is obviously that the number of positive LNs identified is dependent on the total number of LNs resected for pathological examination. I suggest adding a comment. This has been mentioned in lines 211-216.
Results:
Lines 100-103: points out that in the group of 283 pts diagnosed with carcinosarcoma, only in 1/3 (93 pts) the available data allowed the calculation of LNR. The group excluded from further analysis most likely included patients who did not have any lymph node removed. A short comment seems appropriate at this point. This has been further described in lines 149-150 ('the exclusion was due to LND not being perfomed, only 1 LN being resected or because the number of resected nodes were not provided').
​In Table 1, a relatively important group of 16/93 pts with no information about the completeness of the surgical procedure and 28/93 about LVSI is noteworthy. How does this relate to paragraph 1 of the Material and Methods section, which states that patients with incomplete data were excluded from the analysis (line 68-69)? Thank you for this pertinent comment. The exclusion mentioned in this paragraph relates to data regarding recurrnece or survival of the patient. Otherwise patients were not excluded.
Results:
Lines 118-120: …”In multivariate analysis, tumor size remained a significant predictor of PFS (P=0.045) and OS (P=0.38); in addition, 119 positive margin was also a predictor of PFS (P=0.015) (Tables 2 and 3)” - I propose to supplement table 1 with data on pts with positive margins. Judging by the number of pts: Complete resection (71) + Minimal residual disease (<1cm) (4) +Gross residual disease (>1cm) (2) + Not available (16) = 93, then pts with a positive margin have been included here somewhere ( probably to minimal residual disease). Why was this subgroup (positive margin) separated later? Thanks again for this comment. This is due to microscopically positive margins rather than visible residuel disease.
Round 2
Reviewer 1 Report
Comments and Suggestions for Authors
Thanks the authors for the corrections.